# Synergistic Benefits of Using Expansive and Shrinkage Reducing Admixture on High-Performance Concrete

**DOI:** 10.3390/ma11122514

**Published:** 2018-12-11

**Authors:** Tian-Feng Yuan, Seong-Kyum Kim, Kyung-Teak Koh, Young-Soo Yoon

**Affiliations:** 1School of Civil, Environmental and Architectural Engineering, Korea University, 145 Anam-Ro, Seongbuk-gu, Seoul 02841, Korea; yuantianfeng@korea.ac.kr; 2Research Institute for Mega Construction, Korea University, 145 Anam-ro, Seongbuk-gu, Seoul 02841, Korea; envylife@korea.ac.kr; 3Structural Engineering Research Division, Korea Institute of Construction Technology, Deahwa-dong, 283, Goyangdea-ro, Ilsanseo-gu, Goyang-si, Gyonggi-do 10223, Korea; ktgo@kict.re.kr

**Keywords:** high-performance concrete, expansive admixture, shrinkage reducing admixture, autogenous shrinkage, synergistic effect

## Abstract

High-performance concrete (HPC) is widely used in construction according to great mechanical properties, but it has a high risk of shrinkage cracking due to autogenous shrinkage stress. Therefore, the aim of this research was to investigate the effect of a combination of expansive admixture (EA) and shrinkage reducing admixture (SA) on the autogenous shrinkage of high-performance concrete without heat treatment. Two different EA to cement weight ratios of 0.0, 5.0%, and two different SA to cement weight ratios of 0.0, and 1.0% were combined and considered. To investigate the differences in the time-zero conditions effect on the autogenous shrinkage behaviors, four different initial points were compared. The test results indicate that the EA and/or SA content was conductive to a little bite increase compressive strength (22.6–37.9%) and tensile strength (<4.8%). According to the synergistic effect of the EA and SA on the HPC, the autogenous shrinkage significantly decreased (<50%), as compared to those specimens with only one type of admixture (EA or SA). Furthermore, all the specimens incurred restrained autogenous shrinkage cracks at an early age, except the specimen using the combined EA and SA. Therefore, it can be concluded that the combination of EA and SA is effective for improving the properties of HPC.

## 1. Introduction

High-performance concrete (HPC) is widely used in construction given its great mechanical properties, durability, and economic efficiency. Nevertheless, due to a low water-to-binder ratio (*w/b*), extremely high autogenous shrinkage occurs and is vulnerable to evoking shrinkage cracking. The cracking leads to decreased durability in concrete structures that are adequately degenerated due to the ingress of chloride ions. Hence, the autogenous shrinkage behavior of HPC should be exactly evaluated and countermeasures need to be prepared to reduce shrinkage [1,2].

Such this autogenous shrinkage stresses are caused by a loss of moisture to the environment, or external loads or restraints, not as a result of thermal causes. The shrinkage caused by thermal conditions is not from external influences, but it is the result of self-desiccation resulting from chemical shrinkage [3,4,5,6]. In other words, autogenous shrinkage occurs in low *w/b* and the capillary stress in the concrete is considered to be the main shrinkage mechanism. This is because the hydration of the concrete expends water and the consequent relative humidity decrease leads to capillary stresses [4,7].

Several methods for reducing shrinkage are available, such as the use of a surface treatment, expansive admixture (EA) and/or shrinkage reducing admixture (SA), which results in a significant reduction in drying shrinkage and autogenous shrinkage. EA and SA are the most frequently used admixtures for reducing shrinkage [1,8,9]. The main mechanism of action for EA is reacting with C3A content of Portland cement to promote the production of ettringite during an early age based on a chemical composition including calcium oxide (CaO). In the other words, according to formation of the corresponding hydroxide, the chemical components of EA reacting with water slowly produce an expansion [10]. The use of SA in concrete results in reduced evaporation, internal capillary tension and settlement, and a decrease in crack inducing stresses at the top part of the matrix [11,12,13]. Nonetheless, the concept of using SA was contrasted with the benefits of using EA, the effect of SA was usually evaluated as negative. Some researchers [14,15,16,17] reported that using SA reduced the degree of hydration of the matrix, decreased the strength development, delayed setting times, and reduced strength at an early age.

Several researchers [1,4,6,8,18] combined the use of EA and SA as being more valid in reducing shrinkage strain than the sole use of admixture (EA or SA). It is a highly valid curing treatment for shrinkage compensating concrete according to self-induced stress mitigation [8,19]. Furthermore, just used single admixture of EA or SA cannot absolutely avoid the risk of cracking induced by the shrinkage. However, the combination of EA and SA can enhance the efficiency of a shrinkage reduction base via a synergistic effect. However, these are generally evaluated effects of combined EA and SA in autogenous shrinkage under free conditions, and less evaluated the restrained shrinkage behavior. Additionally, some researchers [6,20] just considered a combination of using EA and SA in the autogenous shrinkage test under free and restrained conditions, which it neglects, as compared with using a single admixture.

Therefore, in this study, the evaluate effect of using EA and/or SA on the mechanical properties (setting time, compressive, and tensile strength) and autogenous shrinkage (under free and restrained conditions) of high-performance concrete (> 100 MPa) without heat treatment was investigated using a shrinkage test prepared based on the recommendation of the Japan Concrete Institute (JCI) [21]. Furthermore, to better investigate the correct shrinkage behavior, four different time-zero conditions were designated.

## 2. Experimental Program

### 2.1. Mixture Proportions and Materials

The mixture proportions of the HPC are listed in Table 1. Type I Portland cement (C) of a specific surface area of 3413 cm^2^/g and a density of 3.15 g/cm^3^ was used (produced in Seoul of Korea). Zirconium silica fume (Zr) and blast furnace slag (BS) with a specific surface area of 80,000 cm^2^/g, 4250 cm^2^/g, and a density of 2.50 g/cm^3^, 2.90 g/cm^3^, respectively, were used as cementitious materials (Table 2). Zr was adopted to increase the strength of the concrete via the pozzolanic reaction and fill the voids created by the free water in the matrix, as well as to increase the packing density and to improve flow-ability by introducing ball bearings between the large particles [22]. In addition, silica sand was adopted a grain size smaller than 0.5 mm, and silica flour of a diameter of 2 μm including 98% SiO_2_ were adopted in this study. Table 2 shows the chemical and physical properties of these materials. A *w/b* of 21.7% was adopted and two types of high strength smooth steel fibers with a diameter of 0.2 mm and a length of 16.3 mm, and 19.5 mm were used to increase properties of tensile strength and ductility (Table 3). To improve the suitable fluidity, a polycarboxylate superplasticizer (SP) with a density of 1.06 g/cm^3^ was used. Additionally, to evaluate the characteristics of the previously described mixture with EA and/or SA, two different CSA (Calcium Sulfur Aluminate) EA (of a surface area of 3117 cm^2^/g and a density of 2.98 g/cm^3^) to cement weight ratios (0.0% and 5.0%) and two different glycol-based SA (of a density of 1.02 g/cm^3^) to cement weight ratios (0.0% and 1.0%) were considered (Table 1).

### 2.2. Details of the Experimental Test Setup and Specimen Preparation

#### 2.2.1. Flow and Setting Properties

To evaluate the workability of every HPC mixtures, a slump flow test was performed in accordance with ASTM C143 [23].

The setting properties (initial and final setting times) were determined using a penetration resistance test as per ASTM C403 after the removal of steel fibers from within the mixture [24]. Three cylindrical plastic molds with a diameter of 150 mm and a height of 160 mm were used. The HPC surface was under 10 mm from the top edge of the mold to provide a surface for the application of liquid paraffin oil that prevents rapid water evaporation and the undesirable effect by dry matrix surface during the penetration resistance test. The paraffin oil had no influence on the cement hydration of the paste [25]. The needle penetrated the paste to a depth of 25 ± 2 mm in 10 s and the clear distance rule of needle impressions was considered. The specimens were tested in a room at a constant temperature of 20 ± 1 °C and a relative humidity (RH) of 60% ± 5%.

#### 2.2.2. Compressive and Tensile Strength

Three cylindrical HPC specimens for each variable mixture were used during the compressive strength tests. Specimens with a diameter of 100 mm and a height of 200 mm were cast for the compressive strength test based on ASTM C39 [26].

The tensile strength of the HPC specimens was evaluated after one day from specimens cast over three, seven, and 28 days, respectively. It was measured according to a direct tensile test (dog-bone test) [27,28] at a monotonic rate of 0.4 mm/min using a maximum capacity of 250 kN trough displacement control. To ensure a centric-loading condition and to avoid secondary flexural stress, the direct tensile test apparatus was used under a pin-fixed end condition. Furthermore, to eliminate the displacement capacity of the dog-bone specimens, a specialized steel frame with two Linear Variable Differential Transformers (LVDTs) were attached to the specimens on the side. The details of the geometry and test setup are shown in Figure 1. Three specimens for each variable were fabricated and cured in a room at a constant temperature of 20 ± 1 °C and an RH of 60% ± 5%.

#### 2.2.3. Autogenous Shrinkage Tests

An autogenous shrinkage test under a free condition was created using three prismatic specimens with a width of 100 mm, a height of 100 mm, and a length of 400 mm according to JCI (Japan Concrete Institute) [21]. To eliminate the frictional force between the mold and the HPC matrix, a Teflon sheet and polyester film were placed on the mold. An embedded strain gage (of nearly zero stiffness) and a thermocouple were placed horizontally in the center of the mold before HPC casting to measure the strain and temperature. After HPC casting, to avoid moisture evaporation, the top surface of each specimen was sealing with a polyester film, as shown in Figure 2a.

A restrained autogenous shrinkage was also casted using three prismatic specimens with a cross section of 100 mm × 100 mm, and a length of 1500 mm according to JCI [21]. The details of the setting process of the restrained autogenous shrinkage are shown in Figure 2b. A deformed steel bar was inserted in the center of the specimen. To afford the same autogenous shrinkage stress at the center of the specimen, the center of the steel bar was sealing with a 300 mm of Teflon sheet to eliminate friction between the steel bar and concrete, and an restrained length of 600 mm was used at both ends to afford full autogenous shrinkage stress. To eliminate the friction between the mold and the HPC matrix, a Teflon sheet and polyester film were used on the mold. Additionally, a steel strain gage and a thermocouple were installed at the center of the steel bar before sealing with a Teflon sheet.

All specimens were measured in a room at a constant temperature of 20 ± 1 °C and an RH of 60% ± 5%. To calculate the pure autogenous shrinkage under free conditions, the thermal effects were compensated from the embedded strain gage measured strain using Equation (1) as follows:*ε_a_* = *ε_m_* − *α*∆*T*(1)
where *ε_a_* is the pure shrinkage strain, *ε_m_* is the shrinkage strain measured by the embedded strain gage, *α* is the coefficient of the thermal expansion of the concrete or steel bar, and ∆*T* is the temperature variation.

## 3. Test Results and Discussion

### 3.1. Properties of Fresh HPC

The average flow values are described in Table 1, which were calculated by averaging the maximum diameter of the flow base on the perpendicular diameter. The fluidity showed similar behavior to that of a previous study, and was slightly increased by adding SA [5,29].

To evaluate the setting times, using regression analysis, Equation (2) was used as follows:
*Log*(*PR*) = *a* + *bLog*(*t*)(2)
where *PR* is the penetration resistance (MPa), *t* is the elapsed time (min), and *a* and *b* are the regression coefficients.

Table 4 summarizes the regression coefficients, the coefficient of determination (*R*^2^), and the initial and final setting times. The development of the penetration resistance of the HPC related to the EA and SA is shown in Figure 3. The initial and final setting times were affected by the specimens using the EA and/or SA. The HPC mixtures without the EA and SA showed longer initial and final setting times of approximately 12.06 h and 14.08 h, respectively. The increase in the penetration resistance (initial and final setting times) was delayed as a function of adding SA (0.0E-1.0R and 5.0E-1.0R). This is because using SA decreases the interparticle force leading to reduced water surface tension [11]. However, specimens with the EA (5.0E-0.0R and 5.0E-1.0R) showed a slightly earlier time of initial and final setting times compared to those of the others, which were approximately 31.9–37.9% and 22.6–34.9%, respectively. According to the chemical composition of the CSA EA including the calcium oxide (CaO), it reacted with the C_3_A content of the Portland cement to promote the production of ettringite at an early age, and hence increased the setting times [12,13].

### 3.2. Properties of Hardened HPC

#### 3.2.1. Compressive Strength

Compressive strengths were tested according to the plan age and the results are shown in Table 4. In the case of 0.0E-0.0R, the compressive strength was 103.6 MPa and was of a lower strength than that of the other specimens. The compressive strength increased with increasing use of the type of admixtures. The HPC mixture including SA at 1.0% provided a 4.8% higher compressive strength than that of specimen 0.0E-0.0R. According to those previously reported [1,30], due to using SA decreasing the shrinkage strain, the micro-cracks of the matrix are reduced. The HPC mixture with 5.0% EA (5.0E-0.0R, 5.0E-1.0R) showed a higher compressive strength than that of the 0.0E-0.0R specimen, which were approximately 9.5% and 12.0%, respectively. A possible explanation of this observation is that a marked increase in the volume of the matrix base on the secondary ettringite, which could fill the capillary pores, could basically result in a denser matrix [1,8]. Therefore, the HPC mixture using the combined EA and SA provided a higher compressive strength than that of the others.

#### 3.2.2. Tensile Strength

Table 5 shows the tensile strength results for the HPC with various EA and SA contents. The measurements of tensile strength were taken from one day to 28 days using the dog-bone test method. The tensile strength was markedly increased when increased using admixture contents. The 0.0E-0.0R specimen showed the lowest tensile strength at one, three, seven, and 28 days compared to that of the other specimens at 4.98, 5.30, 6.57, and 6.35 MPa, respectively. The specimen with 1.0% SA showed an improvement in the test age strength of approximately 1.2–23.2% compared to that of specimen 0.0E-0.0R, which is similar to the compressive strength behavior and the improvement in the strength of high strength concrete by using SA was reported by previous researchers [1,8,28,31]. Additionally, the specimen with 5.0% EA showed an improved tensile strength of approximately 2.8–31.5% than specimen 0.0E-0.0R on test ages. It can be expected that the EA lead to a concrete increase of the ettringite and less C-S-H, as compared to the 0.0E-0.0R, which significantly increased strength [8,12,13]. Furthermore, according to the positive properties previously mentioned, the specimens with 5.0% EA also improved tensile strength approximately 1.6–12.0% than specimen with only 1.0% SA. Due to the combined effect of EA and SA, the specimen 5.0E-1.0R exhibited the greatest tensile strength on test ages, which exhibited 6.24, 8.15, 8.79, and 9.25 MPa, respectively.

Furthermore, to analyze the potential properties of shrinkage cracking, the tensile strength development of the HPC with age was necessary. Figure 4 shows the tensile strength development of the HPC with EA and/or SA contents, using the predictive Equation (3) and according to the degree hydration model suggested by previous researchers [32,33]. Equation (3) involves the 28 days tensile strength, similar to Graybeal’s equation, and the predicted behavior similar to the experimental results, as follows:*f_t_*(*t*) = *f_t_*_28_*exp*{−*β*[ln(1 + (*t* − *t*_0_))]*^−k^*}(3)
where *f_t_*_28_ is the tensile strength after 28 days, *β* and *k* are the regression coefficients, and *t*_0_ is the time-zero of the shrinkage behavior.

### 3.3. Autogenous Shrinkage Behavior under Free Conditions

#### 3.3.1. Comparison of the Shrinkage Behavior under Different Time-Zero Conditions

Autogenous shrinkage strains of HPC were recorded from the time of casting. According to those previously reported [34], the maximum values of shrinkage strains can be seriously different depending on the time-zero conditions (shrinkage strains are evaluated from the start point) as the fresh concrete is greatly affected by temperature and the ambient environment. Therefore, defining the initial point of autogenous shrinkage measurement (time-zero) is another problem in need of evaluation. JCI recommended that shrinkage strains of concrete evaluated using the initial set as the time-zero point, excluding the volume change when the concrete is still fresh [21]. Whereas, ASTM C1698 [35] and some previous researchers [36,37] have reported the final set as the time-zero. Some researchers [5,6] have reported using the time-zero point of the shrinkage measurement from the initial or final set is unsuitability. Hence, they suggested that the starting point of the shrinkage strain development or the deviation point between the strain and the temperature in concrete can be determined as the time-zero of autogenous shrinkage behaviors. Thus, the selection of an adequate time-zero of autogenous shrinkage evaluation for HPC is particularly important. Therefore, a comparison of autogenous shrinkage behavior under different time-zero conditions was completed.

Figure 5 shows the comparison of 40 day shrinkage behavior under different time-zero conditions. The thermal expansion coefficient of the HPC was adopted with a value of 11 με/°C in this study [38,39]. The highest value of the shrinkage strain occurred by using as a start point of the shrinkage strain development as time-zero. Autogenous shrinkage strains evaluated from the deviation point of strain and temperature showed very similar values to those evaluated from the start point of the shrinkage strains development, similar to previous research [5]. Otherwise, because the autogenous shrinkage strain behavior showed a steep increase at an early age, the autogenous shrinkage strain value significantly decreased when the setting times were defined as time-zero. The defined time-zero of 40 day autogenous shrinkage strain using initial and final setting times was, respectively, 8.3–21.0% and 13.7–40.0% decreased than those evaluated from the start point of the shrinkage strain development. The specimens with an admixture showed a decreased strain value gap between the start point of the shrinkage strain development and setting times to define as time-zero. Therefore, in this study, the start point of the autogenous shrinkage strain development was suggested as time-zero in the HPC.

#### 3.3.2. Comparison of the Shrinkage Behavior of the HPC to Different Admixture Contents

Figure 6 shows the free autogenous shrinkage behavior of HPC evaluated from the start point of the shrinkage strain development as time-zero. Free autogenous shrinkage of HPC showed similar behavior over 40 days, and steeply increased after the defined time-zero and measurement. Otherwise, at an early age, the steeply increasing rate of shrinkage strain value abruptly reduced and expansion behaviors occurred. According to previous reports [5,6], this is because of the chemical shrinkage and the volume contraction from the negative pressure in the internal voids were self-restrained by the hardening of concrete, during which autogenous shrinkage behaviors during at early age occurred.

Free autogenous shrinkage of HPC decreased with an increase in the admixture contents. The 0.0E-0.0R specimen showed the highest shrinkage strain value compared to those of the other specimens during all test periods. The autogenous shrinkage strain of specimens using EA increased at the point of expansion zone and exhibited behavior similar to that of the 0.0E-0.0R specimen. The main reason for this observation was a markedly increase in the volume of the matrix because of the secondary ettringite [1,8]. As the SA decreases the surface tension of the water in the capillary pores, it thus decreases the magnitude of the capillary stress, the specimens using SA were delayed and the increasing rate of shrinkage strain abruptly reduced [40]. In the other words, the slope of shrinkage reduced by using SA, which means the specimens 0.0E-0.0R and 5.0E-1.0R show similar slops of shrinkage development, and the others yet. This was mainly influence by reduce the rate of hydration of matrix and drying of internal water by using SA [41]. Notably, a markedly decrease in autogenous shrinkage strain with using EA and/or SA was observed prior to 20 h. In addition, a specimen using 1.0% SA or 5.0% EA resulted in a 12.6% and 14.0% reduction in autogenous shrinkage after 40 days compared to the 0.0E-0.0R specimen, respectively. Due to the synergistic effect of EA and SA, a 55.7% reduction in autogenous shrinkage strain occurred (Figure 6a).

Figure 6b shows the temperature behavior of the HPC with various admixtures contents. The maximum temperature decreased with the increased use of admixtures, and the time of maximum temperature was reached earlier with the addition of EA. In contrast, the time of maximum temperature was reached and was delayed with the addition of SA.

### 3.4. Autogenous Shrinkage Behavior under Restrained Conditions

There is not observed shrinkage stress at the fresh condition where the matrix has nearly zero stiffness. The autogenous shrinkage strain of concrete and the internal steel bar strain had quite similar values. Hence, the strains occurring during the fresh state period of the concrete were ignored in this study [33]. The thermal dilation influence was necessarily corrected using an appropriate coefficient of thermal expansion, which is the main reason to get pure restrained strains in the steel bar from the concrete shrinkage. The coefficient of thermal expansion of the deformed steel bar was adopted at a value of 11.7 με/°C in this study [42]. Therefore, the strain responses of the steel bar according different admixtures were evaluated and are shown in Figure 7. The strain behavior was similar to that of the autogenous shrinkage under free conditions. The autogenous shrinkage rapidly increased after time-zero, and the specimens with a single type of admixture (0.0E-0.0R, 5.0R-0.0R, and 0.0E-1.0R) occurred an autogenous shrinkage cracking at a very early age. Due to the synergistic effect of the EA and SA (5.0E-1.0R), a marked decrease in autogenous shrinkage strain was observed at a very early age and the overall the shrinkage strain value was reduced, and hence no shrinkage cracking occurred. This can also be attributed to the tensile strength development of the concrete having a greater value than the autogenous shrinkage stress development at each age.

The autogenous shrinkage stress (*σ_cs_*) can be calculated by using an equation, suggested by the JCI committee on autogenous shrinkage [21], which is as follows:
*σ_cs_* = *E_sr_ε_sr_A_sr_*/*A_c_*(4)
where, *E_sr_* is the elastic modulus of the steel bar, *ε_sr_* is the strain obtained in the steel bar excluding the thermal dilation, *A_sr_* is the area of the steel bar, and *A_c_* is the area of the concrete.

Figure 8 shows a comparison of tensile strengths and autogenous shrinkage stresses for using EA and/or SA contents. The autogenous shrinkage stresses calculated using Equation (4) showed high values at an early age, particularly, and the HPC mixtures with a single type of admixture were much higher than the tensile strength. Therefore, restrained shrinkage crack occurred at a very early age. The shrinkage stresses of specimen 5.0E-1.0R gradually increased after a certain point and maintained a steady value, similar to the autogenous shrinkage behavior under a free condition.

## 4. Conclusions

An experimental investigation was performed to evaluate the synergistic effect of EA and SA on high-performance concrete without heat treatment. The mechanical properties of the setting times and strengths were tested at a planned date, and the autogenous shrinkage behavior under free and restrained conditions was evaluated.

Based on the results of this investigation, the following concluding remarks can be made:(1)The HPC mixture including SA showed slightly earlier setting times, and when using EA were approximately 22.6–37.9% earlier than those of the other specimens without EA. For the compressive strength mixture including admixture contents provided 4.8–12% higher compressive strength than those of specimen 0.0E-0.0R. This is because of the micro-cracks in the matrix are decreased by reducing the shrinkage, and the capillary pores are filled, basically resulting in a denser matrix according to the secondary ettringite when using EA and/or SA;(2)The specimen with 1.0% SA, improved test age tensile strength approximately 1.2–23.2%, and specimen with 5.0% EA, improved the tensile strength approximately 2.8–31.5% compared to the 0.0E-0.0R specimen. Due to the synergistic effect of the EA and SA, the 5.0E-1.0R specimens showed the greatest strength for test ages;(3)A comparison of 40 day shrinkage behavior under different time-zero conditions, using initial and final set as time-zeros were respectively 8.3–21.0% and 13.7–40.0% decreased than those evaluated from the start point of the shrinkage strain development. Therefore, the start point of shrinkage strain development was assumed to be time-zero for the free autogenous shrinkage measurements in this study;(4)The specimens using 1.0% SA or 5.0% EA resulted in a 12.6% or 14.0% reduction in autogenous shrinkage after 40 days compared to the 0.0E-0.0R specimen, respectively. The combined effect of the EA and SA resulted in a 55.7% reduction in autogenous shrinkage. Therefore, just combining the EA and SA, restrained autogenous shrinkage cracks do not occur.

## Figures and Tables

**Figure 1 materials-11-02514-f001:**
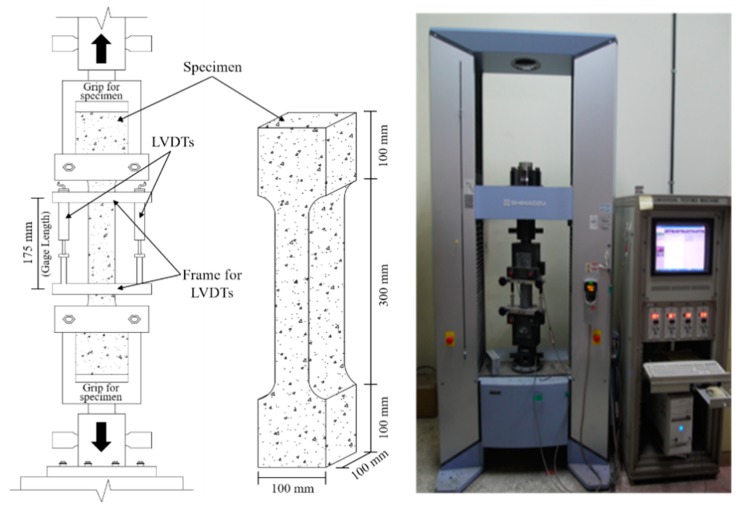
Direct tensile strength test.

**Figure 2 materials-11-02514-f002:**
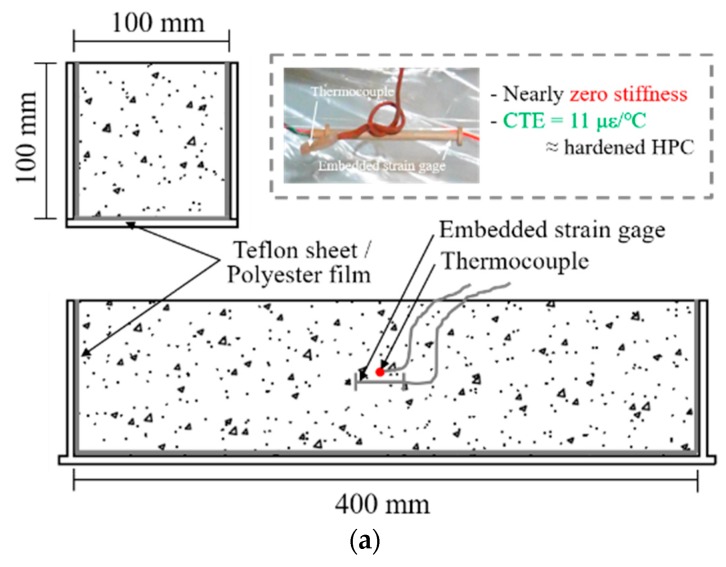
Test setup for autogenous shrinkage strain; (**a**) free condition and (**b**) restrained condition.

**Figure 3 materials-11-02514-f003:**
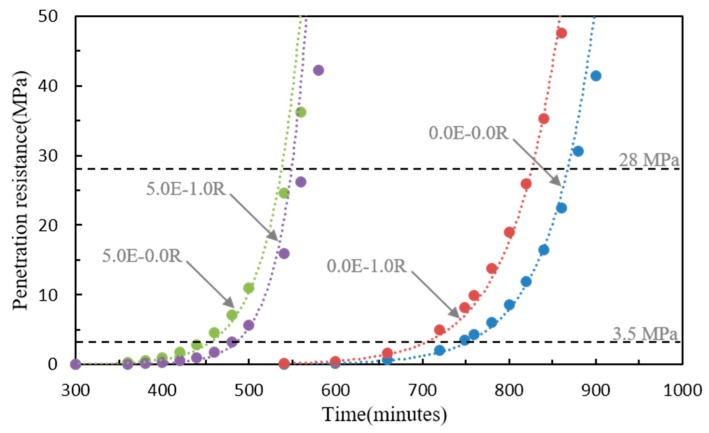
Setting properties of the HPC with various admixture contents.

**Figure 4 materials-11-02514-f004:**
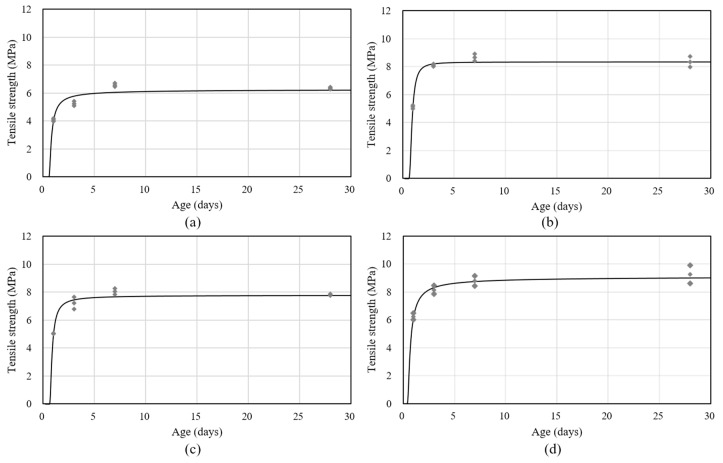
Tensile strength development of the HPC; (**a**) 0.0E-0.0R, (**b**) 5.0E-0.0R, (**c**) 0.0E-1.0R, and (**d**) 5.0E-1.0R.

**Figure 5 materials-11-02514-f005:**
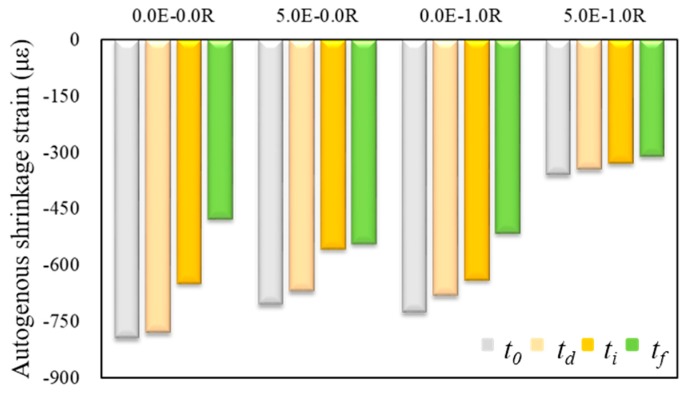
Comparison of 40 day autogenous shrinkage strain evaluated from different start point (start point of shrinkage strain development (*t_0_*), deviation point of shrinkage strain and temperature (*t_d_*), initial setting time (*t_i_*), and final setting time (*t_f_*)).

**Figure 6 materials-11-02514-f006:**
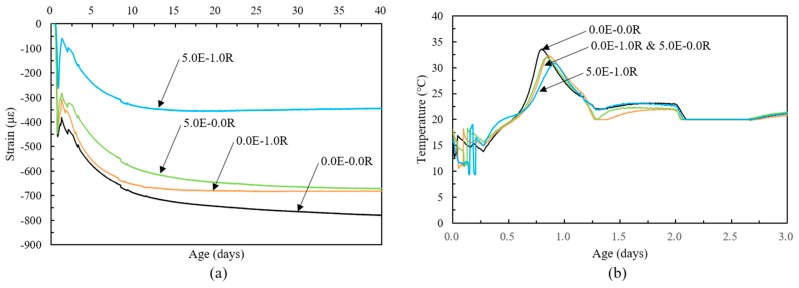
Autogenous shrinkage and temperature behavior; (**a**) autogenous shrinkage and (**b**) temperature.

**Figure 7 materials-11-02514-f007:**
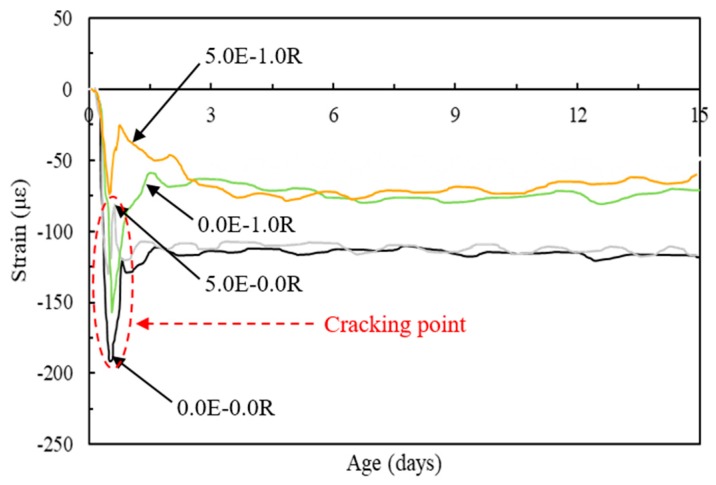
Strain behavior of HPC using different admixtures.

**Figure 8 materials-11-02514-f008:**
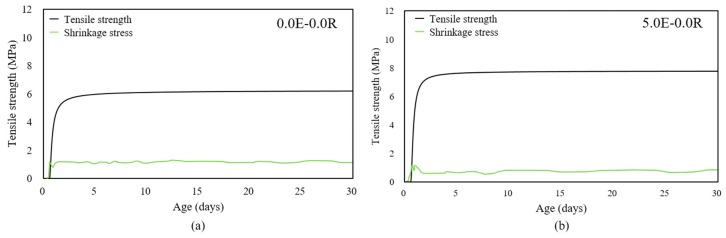
Comparison of tensile strengths and autogenous shrinkage stress; (**a**) 0.0E-0.0R, (**b**) 5.0E-0.0R, (**c**) 0.0E-1.0R, and (**d**) 5.0E-1.0R.

**Table 1 materials-11-02514-t001:** Proportion of materials in the HPC mixture by cement weight ratio.

	*w/b* (%)	C	W	BS	Zr	Silica Sand	Silica Flour	SF (*vf*, %)		SP	EA (%)	SA (%)	Flow (mm)
								19.5	16.3				
0.0E-0.0R	21.7	1.00	0.27	0.18	0.08	1.10	0.30	1.0	0.5	0.02	-	-	820
5.0E-0.0R										0.02	5.0	-	765
0.0E-1.0R										0.01	-	1.0	750
5.0E-1.0R										0.02	5.0	1.0	750

Note: *w/b* = water-to-binder ratio, C = cement, W = water, BS = granulated blast furnace slag, Zr = zirconium silica fume, EA = expansive admixture, and SR = shrinkage reducing admixture.

**Table 2 materials-11-02514-t002:** Chemical compositions and physical properties of cementitious materials.

	Surface Area (cm^2^/g)	Density (g/cm^3^)	Chemical Composition (%)								
			SiO_2_	Al_2_O_3_	Fe_2_O_3_	CaO	MgO	SO_3_	ZrO_2_	Na_2_O	K_2_O
C	3413	3.15	21.25	5.28	3.02	61.00	3.71	1.24	-	-	-
BS	4250	2.90	21.01	6.40	0.09	43.32	5.90	2.78	-	-	-
Zr	80,000	2.50	94.00	0.23	0.10	-	-		3.00	-	-
Silica flour	2.65	0.75	99.60	0.31	0.025	0.010	0.006	-	-	0.009	0.004

Note: C = cement, BS = granulated blast furnace slag, and Zr = zirconium silica fume.

**Table 3 materials-11-02514-t003:** Properties of high strength smooth steel fibers.

Diameter *d_f_* (mm)	Length, *l_f_* (mm)	Aspect Ratio (*d_f_*/*l_f_*)	Density (g/cm^3^)	Tensile Strength (MPa)	Elastic Modulus (GPa)
0.2	16.3	81.5	7.9	2650	200
	19.5	97.5			

**Table 4 materials-11-02514-t004:** Setting properties and compressive strength (*f_ck_*).

Types	*a*	*b*	*R* ^2^	Initial Set (Hours)	Final Set (Hours)	*f_ck_* (MPa)	
						Mean	S.D.
0.0E-0.0R	12.72	35.65	0.973	11.77	13.96	103.65	1.44
5.0E-0.0R	10.57	27.49	0.993	7.49	9.09	110.36	0.32
0.0E-1.0R	13.40	37.97	0.934	12.06	14.08	108.44	1.82
5.0E-1.0R	13.59	35.93	0.938	8.02	9.34	116.12	1.81

Note: *S.D.* = the standard deviation for each value.

**Table 5 materials-11-02514-t005:** Test results of the tensile strength and non-linear regression coefficients.

Types	Tensile Strength (MPa)				Non-Linear Regression Coefficients		
	1-day	3-day	7-day	28-day	*β*	*k*	*R* ^2^
0.0E-0.0R	4.98	5.30	6.57	6.35	0.130	1.402	0.965
5.0E-0.0R	5.12	8.10	8.68	8.35	0.046	3.665	0.982
0.0E-1.0R	5.04	7.23	8.05	7.82	0.074	1.980	0.966
5.0E-1.0R	6.24	8.15	8.79	9.25	0.156	1.449	0.983

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
