# Peer review of "Synergistic Benefits of Using Expansive and Shrinkage Reducing Admixture on High-Performance Concrete"

_materials, 2018, doi:10.3390/ma11122514_

Round 1
Reviewer 1 Report
This paper deals with the synergitic effects of Expansive and shrinkage reducing admixtures to reduce the shrinkage of High performance concrete.
The paper is well written and the authors aim is sound.
I feel that the paper should be accpeted for publication.
I have just one question: could the author estimate the entrapped air in order to explain the differences in mechanical strength?
Author Response
This paper deals with the synergistic effects of expansive and shrinkage reducing admixtures to reduce the shrinkage of High performance concrete.
The paper is well written and the authors aim is sound.
I feel that the paper should be accepted for publication.
Answer: First of all, thank you very much for your useful comments on our paper. We would like to thank you for your excellent comments which significantly improved the quality of our paper.
1) I have just one question: could the author estimate the entrapped air in order to explain the differences in mechanical strength?
Answer: In general, the concrete matrix using expansive admixture (EA) shows a high strength properties than those of specimen without EA. A possible explanation of this observation is that a significant increase in the volume of the concrete matrix due to the secondary ettringite, which could fill the capillary pores and basically result in a denser matrix [1, 2]. Therefore, we have ignored to estimate the entrapped air of specimens in this study. Thank you very much for your comments, we will consider and estimate the entrapped air in the further research.
1. Meddah M.S.; Suzuki M.; Sato R. Influence of a combination of expansive and shrinkage-reducing admixture on autogenous deformation and self-stress of silica fume high-performance concrete. Constr. Buil. Mater. 2011, 25, 239-250.
2. Meddah M.S.; Suzuki M.; Sato R. Combined effect of shrinkage reducing and expansive agents on autogenous deformation of high-performance concrete, Sustainable concrete technology and structures in local climate and development conditions. In: Proceeding of the 3rd ACF international conference – ACF/VCA-2008. HoChiMinh City, Vietnam; November 11-13, 2008. p. 339-46.
Reviewer 2 Report
The authors studied the effect of combining expansive admixture and shrinkage reducing admixture on the autogenous shrinkage of high-performance concrete without heat treatment. The manuscript is within the scope of Materials Journal. However, before publication some major concerns must be addressed:
The introduction must be updated since in the present form does not reflect the actual state of the art.
Considering the heterogeneity of concrete please detail how many samples were studied for each test and clearly indicate the standard deviation for each test.
Author Response
The authors studied the effect of combining expansive admixture and shrinkage reducing admixture on the autogenous shrinkage of high-performance concrete without heat treatment. The manuscript is within the scope of Materials Journal. However, before publication some major concerns must be addressed:
Answer: First of all, thank you very much for your useful comments on our paper. We have carefully considered all your comments, and the revised manuscript is now attached for your reconsideration. We really appreciate the opportunity to resubmit. Also, we would like to thank you for your excellent comments which significantly improved the quality of our paper.
1) The introduction must be updated since in the present form does not reflect the actual state of the art.
Answer: As you recommended, the introduction is updated.
There are 17 references to introduce the background, necessity of research, and current situation by using EA and SA, that references has symbolic significance. And there is upload the actual state of the art in the introduction.
According to several researchers [1, 4, 6, 8, 13], the combined use of EA and SA is more efficient in reducing shrinkage than the sole use of single admixture (EA or SA). It is efficient curing treatment for shrinkage compensating concrete according to self-induced stress mitigation [8, 14, 15]. But there are generally evaluated effect of combined using EA and SA in autogenous shrinkage under free condition, and less of evaluated the restrained shrinkage behavior. Furthermore, just used single admixture of EA or SA cannot absolutely avoid the risk of cracking induced by shrinkage. However, the combined use of EA and SA can enhance the efficiency of a shrinkage reduction base via a synergistic effect.
2) Considering the heterogeneity of concrete please detail how many samples were studied for each test and clearly indicate the standard deviation for each test.
Answer: The compressive strength test and specimens fabricated were based on ASTM C403. The tensile strength test and specimens were based on the preview researches. And all of these researchers using three specimens of all each variation. The autogenous shrinkage was also according to JCI, and the number of specimens should use minimum three. Furthermore, in Europe, The European standard EN 206 defines the number of samples to be tested. The recommended that should use minimum 3 cubes or cylinders. Hence In this study, three samples for each variation were fabricated and tested.
Reviewer 3 Report
Title:
ok
Abstract:
It is requested to indicate the need to carry out this investigation.
It is requested to indicate the most important results in specific values or percentages.
It is requested to expose the practical advantages of the exposed results.
Keywords:
Ok
Introduction
It is considered that the information provided in the state of knowledge is limited. It is requested to increase the bibliographical references.
There are several aspects that should be discussed: a / c ratio, concrete surface area, environmental conditions of temperature and relative humidity, shape effect of the specimen, surface effect of the specimen, porosity and its tortuosity, etc.
Mix of proportions and materials:
Indicate compliance with regulations (for example, ASTM for all components used)
Establish the dosage method used (normative) and the criteria to establish the quantities of other materials of the mixtures.
Verify that the regulations (for example, ASTM) that were used in all the tests are indicated.
Indicate the procedure for mixing the samples and the total quantities of each mixture used.
Resistance to compression and traction:
It is necessary to indicate all the laboratory equipment used (brand, model and country of origin)
Figure 2a, improve the quality of the image
Compressive strength:
It is recommended to simplify the information (tables and figures), it is recommended to only leave the figures. Apply the criterion in all the work.
Tensile strength:
Figure 5. What is td ?; indicate the units of the axes
Authors are recommended to perform XRD tests that validate the hypotheses of shrinkage behavior attributable to the formation of chemical compounds. The previous requirement is mandatory.
Figure 8. Is it possible to perform a more detailed analysis of the implications between resistance and shrinkage?
Conclusions
The authors are asked to update the conclusions after making the previous improvements.
Bibliography:
Authors are asked to improve the number of references. It is recommended to use SCOPUS, Compendex or WoS.
Author Response
First of all, thank you very much for your useful comments on our paper. We have carefully considered all your comments, and the revised manuscript is now attached for your reconsideration. We really appreciate the opportunity to resubmit. Also, we would like to thank you for your excellent comments which significantly improved the quality of our paper.
1) Abstract: It is requested to indicate the need to carry out this investigation. It is requested to indicate the most important results in specific values or percentage. It is requested to expose the practical advantages of exposed results.
Answer: As you recommended, the abstract is updated.
2) It is considered that the information provided in the state of knowledge is limited. It is requested to increase the bibliographical references. There are several aspects that should be discussed: a/c ratio, concrete surface area, environmental conditions of temperature and relative humidity, shape effect of the specimen, surface effect the specimen, porosity and its tortuosity, etc.
Answer: As you recommended, the references are updated and total using 37 references to discussion which has symbolic significance.
The results and discussion considered effect of admixture type, chemical properties, and porosity, etc. Because all specimens investigated the strength and autogenous shrinkage under same specimens size. And all curing in a room at a constant temperature of 20±1 °C and an RH of 60±5% until test age.
3) Indicate compliance with regulations (for example, ASTM for all components used). Establish the dosage method used (normative) and the criteria to establish the quantities of other materials of the mixtures. Verify that the regulations (for example, ASTM) that were used in all the tests are indicated. Indicate the procedure for mixing the samples and the total quantities of each mixture used.
Answer: As you recommended, the described more details of materials.
Type 1 Portland cement (C) of a specific surface area of 3,413 cm2/g and a density of 3.15 g/cm3 was used (produced in Korea). Zirconium silica fume (Zr) and blast furnace slag (BS) with a specific surface area of 80,000 cm2/g, 4,250 cm2/g and a density of 2.50 g/cm3, 2.90 g/cm3, respectively, were used as cementitious materials. And the Chemical composition and physical properties were described in Table 2, and Table 3.
According to previous researchers, the usually described the mixture proportion by weight ratio. Because of projects confidentiality agreement.
And for the procedure for mixing is similar UHPC. Firstly, cement, blast furnace slag, silica fume, silica flour, and sand were premixed for about 10 min without addition of water. Water and superplasticizer (some admixture) were applied and mixed for about 10 min thereafter. Confirming that the dispersion of paste is adequate and the mixture become flowable, the steel fibers were added by using fiber feeder, which provide enough scatteration of fiber with vibrating system, and then mixed for an additional 5 min.
4) Resistance to compression and traction: It is necessary to indicate all the laboratory equipment used (brand, model and country of origin).
Answer: All apparatus regular and available on the international or business company. The U.T.M was made in Germany, embedded and steel gage, thermocouple, and LVDTs was made in Japan.
5) Figure 2a, improve the quality of the image.
Answer: As you recommended, image was changed.
6) Compressive strength: It is recommended to simplify the information (table and figures), it is recommended to only leave the figure. Apply the criterion in all the work.
Answer: Thank you for your recommendation, but other reviewers suggest give out the tables to confirm the figures.
7) Tensile strength: Figure 5. What is td? Indicate the unites of axes. Authors are recommended to perform XRD test that validate the hypotheses of shrinkage behavior attributable to the formation of chemical compounds. The previous requirement is mandatory.
Answer: As you recommended, Fig. 5 is exchanged.
And start point of shrinkage strain development (t0), deviation point of shrinkage strain and temperature (td), initial setting time (ti), final setting time (tf). Furthermore, thank you for your recommended, we will perform XRD test in the further research.
8) Figure 8 is it possible perform a more detailed analysis of the implications between resistance and shrinkage?
Answer: Fig. 8 shows a comparison of tensile strengths and autogenous shrinkage stresses for various EA and SRA contents. The tensile strength development was calculated using an equation, suggested by the JCI committee on autogenous shrinkage [14]; as follows:
σcs = EsrεsrAsr / Ac (4)
where, Esr is the elastic modulus of the steel rebar, εsr is the strain obtained in the steel rebar excluding the thermal dilation, Asr is the area of the steel rebar, and Ac is the area of the concrete.
According this tensile strength development we just certify the shrinkage cracking was occurred.
Therefore, the autogenous shrinkage stresses calculated using equation showed high values at an early age, particularly, and the HPC mixtures with a single type of admixture were much higher than the tensile strength. Therefore, autogenous restrained shrinkage crack occurred at a very early age. The shrinkage stresses of specimen 5.0E-1.0R gradually increased after a certain point and maintained a steady value, similar to the autogenous shrinkage behavior under a free condition.
9) Conclusions: the authors are asked to update the conclusions after making the precious improvement.
Answer: As you recommended, update the conclusions.
Reviewer 4 Report
This manuscript presents the synergistic effect of expansive admixture (EA) and shrinkage admixture (SRA) on autogenous shrinkage of high performance of concrete (HPC). This work could be interesting for industrial applications, but still needs to deeper scientific discussion on the coupled effects of 2 admixtures. Main issues should be reconsidered as follows:
As superplasticizer was added into HPC, the ternary effect (SP+EA+SRA) should be investigated and further discussed.
A wider range (weight ratio) of EA and SRA should be investigated. Strictly speaking, 0% cannot considered as and investigated weight fraction, but just a reference point.
Introduction should be rewritten: please explain how thermal shrinkage is linked to self-desiccation (lines 39-40). What is the novelty of this research compared to the ones in references [8] and [13].
Many results have been shown, but critical discussion is missing.
Microstructure of the specimens should be examined to confirm the formation of secondary ettringite and detect any micro crack occurred.
English should be rechecked through the manuscript.
The reviewer suggest a major revision before further consideration for publication.
Author Response
This manuscript presents the synergistic effect of expansive admixture (EA) and shrinkage admixture (SRA) on autogenous shrinkage of high performance of concrete (HPC). This work could be interesting for industrial applications, but still needs to deeper scientific discussion on the coupled effects of 2 admixtures. Main issues should be reconsidered as follows:
Answer: First of all, thank you very much for your useful comments on our paper. We have carefully considered all your comments, and the revised manuscript is now attached for your reconsideration. We really appreciate the opportunity to resubmit. Also, we would like to thank you for your excellent comments which significantly improved the quality of our paper.
1) As superplasticizer was added into HPC, the ternary effect (SP+EA+SRA) should be investigated and further discussed.
Answer: As you recommended, the ternary effect should be interesting topic. Therefore, we would like to thank you for your recommendation and we will consider it in the further research.
2) A wider range (weight ratio) of EA and SRA should be investigated. Strictly speaking, 0% cannot considered as and investigated weight fraction, but just a reference point.
Answer: The test variables were consider and decision based on the previous research. For example, Yoo et. Al (2015) published using 1.0% of SRA can significantly reducing shrinkage strain and prevent restrained shrinkage cracking. But Tanimura et. al (2002), Sato et. al (1999) have reported that the use of combination of SRA and EA as well as a ternary blended system is more efficient to reduce shrinkage than the single-incorporation of either SRA or EA separately. The aims of this study is investigate the effect of a combination of EA and SRA on the autogenous shrinkage of HPC, and find out no shrinkage cracking mixture. Hence, just considered 0.0E-0.0R, 5.0E-0.0R, 1.0E-0.0R, 5.0E-1.0R. And we already consider and defined the 0.0% as a reference point. Furthermore, we would like to think wider range of admixture in the further research.
1. Yoo D.Y.; Banthia N.; Yoon Y.S. Effectiveness of shrinkage reducing admixture in reducing autogenous shrinkage stress of ultra-high-performance fiber-reinforced concrete, Cem. Concr. Comps. 2015, 64, 27-36.
2. Tanimura M.; Hiramatsu Y.; Hyodo H.; Sato R. Flexural performance of RC members made of low shrinkage high performance concrete. In: Proceeding of the 6th international symposium on utilization of high strength/high performance concrete. Leipzig, Germany; 2002, 2, 1437–52.
3. Sato R, Tanaka S, Hayakawa T, Tanimura M. Experimental studies on reduction of autogenous shrinkage and its induced stress in high-strength concrete, autogenous shrinkage of concrete. In: Tazawa EI, editor. Proceeding of the international workshop. London: Japan Concrete Institute, E & FN Spon; 1999. p. 163–71.
3) Introduction should be rewritten: please explain how thermal shrinkage is linked to self-desiccation (lines 39-40). What is the novelty of this research compared to the ones in references [8] and [13].
Answer: Meddah et. al (2008, 2009, 2010,) report combined effect of shrinkage reducing expansive agents on autogenous deformations of high-performance concrete, Self-induced stress in high performance concrete treated with a combination of shrinkage-reducing and expansive admixture, and effect of curing methods on autogenous shrinkage and self-induced stress of high performance concrete. The self-induced tensil stress developed over time under sever conditions. On the other hand, SF-HPC is known to exhibit a higher magnitude of autogenous shrinkage compared to mixtures containing other types of pozzolanic materials such as fly ash or slag. Therefore, the effectiveness of each curing system that might be achieved under such conditions (very low w/b and HPC with SF) should be more efficient and could easily be validated in conventional and practical conditions.
4). Microstructure of the specimens should be examined to confirm the formation of secondary ettringite and detect any micro crack occurred.
Answer: The microstructure of the phenomena was followed reference, which using some kinds of admixture. The XRD, and XRF of the specimens will consider in the further research as you recommended.
Round 2
Reviewer 2 Report
All the concerns were well addressed. Therefore, I support publication of the manuscript after English review.
Reviewer 3 Report
Some improvements have been made; however, 2 requirements persist (they are mandatory and have not been carried out). Both requirements are feasible to perform and the authors are asked to include them again. Without these two requirements, the work can not be published.
Requirement number 2
Requirement number 7
Reviewer 4 Report
Not much improved compared to previous version except for language. I would suggest the authors to resubmit again once the further suggested analyses/examinations are performed.
